# Step by Step through the Years—High vs. Low Energy Lead Extraction Using Advanced Extraction Techniques

**DOI:** 10.3390/jcm11164884

**Published:** 2022-08-19

**Authors:** David Zweiker, Basma El Sawaf, Giuseppe D’Angelo, Andrea Radinovic, Alessandra Marzi, Luca R. Limite, Antonio Frontera, Gabriele Paglino, Michael Spartalis, Donah Zachariah, Kenzaburo Nakajima, Paolo Della Bella, Patrizio Mazzone

**Affiliations:** 1Department of Cardiac Electrophysiology and Arrhythmology, IRCCS San Raffaele Scientific Institute, Vita-Salute University and San Raffaele Hospital, Via Olgettina 60, 20132 Milan, Italy; 2Third Clinical Department for Cardiology and Intensive Care, Pavillon 29, Klinik Ottakring, Montleartstraße 37, 1160 Vienna, Austria; 3Division of Cardiology, Medical University of Graz, Auenbruggerplatz 15, 8036 Graz, Austria; 4Department of Electrophysiology and Electrostimulation, Humanitas Research Hospital, Via Manzoni 56, 20089 Milan, Italy

**Keywords:** transcatheter lead extraction, implantable cardiac defibrillator, retrospective analysis, risk score

## Abstract

Background: Limited data is available about the outcome of TLE in patients with vs. without high energy leads in the last decade. Methods: This is an analysis of consecutive patients undergoing TLE at a high-volume TLE centre from 2001 to 2021 using the stepwise approach. Baseline characteristics, procedural details and outcome of patients with high energy lead (ICD group) vs. without high energy lead (non-ICD group) were compared. Results: Out of 667 extractions, 991 leads were extracted in 405 procedures (60.7%) in the ICD group and 439 leads in 262 procedures (39.3%) in the non-ICD group. ICD patients were significantly younger (median 67 vs. 74 years) and were significantly less often female (18.1% vs. 27.7%, *p* < 0.005 for both). Advanced extraction tools were used significantly more often in the ICD group (73.2% vs. 37.5%, *p* < 0.001), but there were no significant differences in the successful removal (98.8% vs. 99.2%) or complications (4.7% vs. 3.1%) between the groups (*p* > 0.2 for both). Discussion: Using the stepwise approach, overall procedural success was high and complication rate was low in a high-volume centre. In patients with a high energy lead, the TLE procedure was more complex, but outcome was similar to comparator patients.

## 1. Introduction

In recent years, there have been increasing numbers of cardiac implantable electric device (CIED) implantations [1]. While the CIED implantation is considered a low-risk procedure [2], serious complications, such as infection, sometimes require transvenous lead removal [3,4]. The longer the lead has been implanted, the higher is the risk of adhesions with the vasculature and the heart [5]. These adhesions may lead to several life-threatening complications, such as venous rupture, cardiac tamponade and arrhythmias [4]. Transvenous lead extraction (TLE) involves the removal of leads with a dwell time of at least one year and is considered a high-risk procedure requiring adequate training and backup cardiac surgery ready [6].

In high-energy (ICD) leads, the high lead surface area and the presence of coils may facilitate infection and adhesions [5] leading to more complex TLE procedures. Therefore, patients with an ICD represent a special subpopulation. Furthermore, alternatives to traditional transvenous devices, such as subcutaneous defibrillators, are available to a proportion of those patients [7]. We hypothesized that TLE of high energy leads may be associated with increased complexity and worse outcome. We therefore performed an analysis of baseline, procedural and outcome characteristics in patients with high energy vs. low energy (pacemaker) leads in the last 20 years, in whom TLE was necessary. Furthermore, we evaluated the validity of two established risk scores for predicting procedural complexity and complications.

## 2. Materials and Methods

The Department of Cardiac Electrophysiology and Arrhythmology, San Raffaele Hospital, Milan, Italy serves as an international reference centre for various procedures in electrophysiology, including TLE and ventricular tachycardia ablation, performing over 2000 procedures per year, including 40–60 TLE procedures. This is a retrospective analysis of all patients undergoing TLE at this centre from 3 September 2001 to 2 September 2021. The institutional review board (Vita-Salute University San Raffaele, Milan, Italy) approved the study.

### 2.1. Stepwise Approach for TLE

TLE procedures were performed by experienced electrophysiologists in the electrophysiology laboratory with a cardiac surgery team ready. Indication for TLE was made according to local and international guidelines [4]. All patients underwent invasive blood pressure monitoring and were either sedated or anesthetized and intubated for the procedure, depending on individual circumstances. In case of infection, transoesophageal echocardiography was performed to diagnose or exclude vegetations on the leads or cardiac valves. To intervene in case of lead rupture, two sheaths were inserted into the femoral vein. Furthermore, a temporary pacing lead was introduced via one of the femoral sheaths. Afterwards, the CIED pocket was opened, and the leads were prepared for extraction.

The institution developed a standard operating procedure to optimize the outcome of TLE. Details have been published elsewhere [8]. In short, manual traction with regular and locking stylets (LLD, Spectranetics, Colorado Springs, CO, USA, or Liberator, Cook Medical, Bloomington, IN, USA) was performed. If unsuccessful, telescoping with non-powered dilator sheaths was tried (Byrd dilator, Spectranetics). Advanced extraction methods were used as a next step, including mechanically powered sheaths (Evolution RL, Cook Medical; TightRail, Spectranetics) or laser sheaths (SLS II Laser Sheath, Spectranetics). Depending on the clinical situation, non-powered dilator sheaths were used along with powered sheaths, especially together with Evolution sheaths. In case of lead rupture or on discretion of the operator, a snare (Needle’s-eye snare, Cook Medical) was inserted via the femoral route to retrieve remaining parts.

Depending on the clinical condition after the procedure, patients were monitored in the recovery room for at least one hour or transferred to the intensive care unit. Further clinical management depended on the indication for TLE. If infection was excluded and an indication for CIED implantation was still valid at the time of TLE, the device implantation was performed immediately after TLE.

### 2.2. Outcome Definitions

Outcomes were defined according to the 2017 Heart Rhythm Society consensus statement [4]. Complete retrieval was defined as the complete removal of the whole CIED lead. Clinical success was the primary endpoint and was defined as removal of all targeted leads completely or retention of a small proportion of the lead with a low risk of further complications (e.g., the tip of the lead in the endocardium). Secondary endpoints were the use of advanced extraction tools and the occurrence of complications. Major complications were defined as either life threatening or resulting in death, significant disability or requiring major surgical intervention to prevent any of the outcomes listed above [9].

### 2.3. Data Collection

All TLE procedures were documented in an institutional database, including details of the patient, the explanted device and short-term outcome. To add details of procedure and fluoroscopy times, procedures were matched with the department’s fluoroscopy database. The merging of both databases was achieved in >94% of cases. Microsoft Excel^®^ (Microsoft Corporation, Redmond, WA, USA) was used for data collection.

### 2.4. Study Groups

Patients were stratified based on the planned extraction of a high energy lead (ICD group). Remaining patients were united into the non-ICD group.

### 2.5. Statistics

Variables were reported as mean ± standard deviation, median (interquartile range) or proportion (absolute number), as appropriate. To examine the temporal trend, procedures were separated into four five-year time intervals (from 3 September 2001 to 2 September 2006, 3 September 2006 to 2 September 2011 and so forth). Chi square and Pearson correlation tests were performed to examine temporal changes in baseline characteristics, procedural details and outcome. To evaluate the previously published MB score [8], we applied it to the whole population to predict the need for complex procedures and complications using receiver operating characteristic (ROC) curves. The MB score is a clinical score consisting of the following 6 variables: Lead age ≥ 3 years ≥ 5 years and ≥10 years; ≥2 leads implanted; presence of one lead with passive fixation and presence of one ICD lead. We evaluated the performance of the SAFeTY TLE score [10] in a similar manner. This score includes the sum of lead dwell times, anaemia, (female) gender, previous procedures and patient age (7). Due to missing clinical information, we excluded the parameter “anaemia”. Furthermore, a bivariate analysis was applied to find predictors for short-term complications. All statistics were performed with R 4.1.2 (The R Foundation for Statistical Computing, Vienna, Austria) using RStudio 2021.09.1 Build 372 (RStudio, Boston, MA, USA).

## 3. Results

During the observation period, a total of 1430 leads were extracted in 667 TLE procedures, which were performed in 655 individual patients. Out of those procedures, 991 leads were extracted in 405 procedures (60.7%) in the ICD group and 439 leads in 262 procedures (39.3%) in the non-ICD group (central illustration).

Twelve patients had two TLE procedures within the observation period. The median age was 70 years (total range 22–94 years) and 22.0% were female. Age was significantly lower (67, IQR 58–76 vs. 74, IQR 63–80 years, *p* < 0.001) and there were significantly fewer women in the ICD group (18.1% vs. 27.7%, *p* = 0.003). There were no significant differences in the prevalence of comorbidities between groups, such as hypertension (ICD 43.7% vs. non-ICD 47.5%), diabetes mellitus (22.9% vs. 19.1%) and chronic kidney disease (17.1% vs. 19.2%, *p* > 0.2 for all).

### 3.1. CIED Details and Indications

Cardiac resynchronisation therapy (CRT) was significantly more prevalent in the ICD group (47.3% vs. non-ICD group 18.3%), while the most common device in the non-ICD group was a PM (74.4%, Table 1). A conduction system lead was present in two cases in the non-ICD group (0.3%). The most common indications for CIED implantation were dilated cardiomyopathy (40.0%) and ischemic cardiomyopathy (37.0%) in the ICD group, and atrioventricular block (38.2%) and sick sinus syndrome (28.2%) in the non-ICD group (*p* < 0.001 between groups). While the ICD group consisted of patients with an implanted ICD (52.6%) or CRT-D (47.4%) only, 12.6% of patients in the non-ICD group had an ICD lead in place that was not extracted.

Out of 1430 implanted leads, 484 had been positioned in the right atrium, 722 in the right-ventricle (RV) and 223 in the coronary sinus (LV, Table 2). Median lead dwell time was 52 (LV, IQR 29–82) to 77 (RV-PM, IQR 34–154) months, with no differences between both groups. Significantly more leads per patient were present in the ICD group (mean ± SD 2.6 ± 0.9 compared to the non-ICD group 2.1 ± 0.6, *p* < 0.001).

Device infection was the leading indication to perform TLE in both groups but was significantly more prevalent in the ICD group (61.7% vs. 44.7%), including pocket infection (37.3% vs. 31.7%) and endocarditis (24.4% vs. 12.6%, *p* < 0.001). Lead dysfunction was significantly more prevalent in the non-ICD group (45.4% vs. 35.6%, *p* = 0.012). Other indications were present in 6.4%.

Details about preprocedural echocardiography and blood cultures were available in 87.3% of cases (*n* = 582). Transoesophageal echocardiography had been performed in 58.7% (*n* = 342), showing vegetations on the leads in 10.5% (*n* = 61). Blood cultures were positive in 57.2% (*n* = 333). Due to missing data in the majority of cases, the distribution of pathogen strains was not analysed.

### 3.2. Outcome

Clinical success as primary outcome was achieved in 98.8% in the ICD group and in 99.2% in the non-ICD group, with no differences between groups (*p* = 0.256). Advanced extraction tools were used significantly more often in the ICD group compared to the non-ICD group (73.2% vs. 37.5%, *p* < 0.001). There were no differences in complications between both groups (4.7% vs. 3.1%, *p* = 0.421).

Median procedure time was 120 min (IQR 105–170), the median fluoroscopy time was 9.5 (5.2–17.2) minutes and the median fluoroscopy dose was 23 (10–51) Gy·cm^2^, with no significant differences between ICD and non-ICD groups (*p* > 0.2 for all).

A maximum of six leads were explanted from a single patient during the same procedure. In all patients, manual traction was performed, using stylets (regular or locking) in 54.3% without differences between groups. Byrd dilator sheaths were used in 2.4%. Complete explant of all leads using manual traction and/or non-powered dilator sheaths was significantly lower in the ICD group compared to the non-ICD group (24.7% vs. 56.9%, *p* < 0.001, Table 3).

Evolution sheaths were used most often (58.3%), followed by SLS II (27.0%) and TightRail (6.1%). These techniques led to a complete retrieval in 84.6% without differences between the two groups. Snares were used in 11.8%, further increasing the complete retrieval rate to 94.8%. One lead was immediately abandoned without any advanced retrieval attempt in the ICD group, and one lead was primarily removed surgically in the non-ICD group. Finally, all leads were successfully retrieved in 95.8% of procedures in the ICD group and in 93.5% in the non-ICD group (*p* = 0.369).

Complete extraction success using various advanced extraction sheaths ranged from 90.2% to 95.0% and were comparable overall and in both groups (Appendix A).

Overall, in-hospital mortality was 0.2%, and major complications occurred in 0.7% without significant differences between groups. Total intraprocedural complications occurred in 2.1% of cases. There were three cases of pericardial tamponade requiring urgent pericardial drainage (two in the ICD-group, 0.5% and one in the non-ICD-group, 0.4%). One patient in the ICD group developed refractory cardiac arrest and died during the procedure. Another patient in the ICD group underwent emergency cardiac surgery due to rupture of the upper vena cava. Further complications were the occurrence of new atrioventricular block (ICD group), pulmonary embolism (ICD group), femoral arteriovenous fistula (ICD group), pneumothorax (non-ICD group), pocket hematoma (ICD group) and migration of the tip of one lead into the liver (non-ICD group; one patient each).

Post-procedural complications were seen in 1.9% of procedures, including haematoma requiring drainage or reintervention (0.6%), revision of the pocket due to other reasons (0.4%). Two patients (0.3%) developed severe tricuspid regurgitation, one patient each developed late pericardial tamponade, pneumothorax and peripheral thromboembolism. One patient died within 24 h after the procedure. There were no differences in the rate of complications or death after the procedure in ICD vs. non-ICD groups.

### 3.3. Predictors of a Complex Procedure and Complications

There was a high correlation between the previously published MB score [8] and the risk for a complex procedure (ROC-area under the curve [AUC] 0.810, *p* < 0.001, Appendix A). Patients with an MB score of zero had no complex procedures, while patients with 6 points had a risk of 97.0%. The modified SAFeTY TLE risk score showed a lower but significant correlation (ROC-AUC 0.676, *p* < 0.001, Appendix A).

We found the number of implanted leads, high lead age and the use of complex extraction tools to be significantly associated with the occurrence of complications. No association between the indication for TLE and complications was found. The MB score and modified SAFeTY TLE score were significantly correlated with the occurrence of complications (MB score: ROC-AUC 0.656, *p* = 0.005; SAFeTY TLE score: ROC-AUC 0.611, *p* = 0.039; Table 4).

### 3.4. Temporal Trend in Procedures

A mean of 33 procedures was performed per year. The rate of extracted leads and TLE procedures in any five-year interval continuously increased from 62 leads in 30 procedures (2001–2006) to 544 leads in 261 procedures (2016–2021). The median age of patients increased from 62 to 71 years (*p* < 0.001, Appendix A). Over the time interval, there was a continuous increase of lead dwell time (median 67 to 79 months), while the number of total leads declined significantly (2.38 to 2.22 leads per procedure, *p* < 0.001 for both).

Over the observation period, procedure duration and fluoroscopy time was similar, while fluoroscopy dose dropped significantly (median 23 to 16 Gy·cm^2^, *p* < 0.001). Relatively frequent use of step two (Byrd dilator, 5.1% vs. 0.0–2.9%) and infrequent use of step three (powered sheaths, 46.8% vs. 60.0–65.5%) in the third quartile compared to remaining quartiles led to significant differences in the use of steps two and three throughout the observation period. Complete retrieval rate of steps one and one-two also varied between the observation quartiles. However, in both outcomes, no clear linear trend was seen. There were no significant changes in overall procedure success or procedural complications.

## 4. Discussion

This large single-centre analysis of consecutive patients undergoing 667 TLE procedures shows that advanced extraction tools were used significantly more often in patients with high energy lead, but success and complication rates were similar compared to remaining patients. Additionally, we found a steadily favourable outcome in the last 20 years despite continuously increasing lead dwell time.

This analysis shows the increasing need of TLE in today’s CIED population, with an increase of procedures by 870% between the first and the last five-year interval. However, the proportion of complex TLE procedures remained similar over the observation period.

Almost 60% of patients undergoing TLE had a device with defibrillation lead implanted, although PM implantations are performed five times as often as ICD implantations [11]. This finding suggests that high energy leads are prone to more complications requiring TLE than regular pacemaker leads, mainly driven by device infection. An indication for ICD implantation according to current guidelines [12] and the evaluation of alternatives to transvenous defibrillation systems [13] is, therefore, crucial, also after TLE [14]. In recent publications of high-volume centres [15,16,17,18,19,20,21,22], baseline characteristics were similar, but the proportion of TLE patients with ICD was lower than in this analysis (58.5%) with a range from 24 [15] to 48% [19]. Infection represented the main indication in most studies [15,16,17,18,19,20]. Apart from significantly lower age, lower prevalence in women, and, obviously, the presence of a high energy lead in the ICD group, the baseline risk characteristics were similar between groups.

The stepwise approach [8,23], which is favoured in our centre, allows the use of advanced extraction tools while minimizing risk. It may be especially helpful for operators with limited experience. This analysis clearly shows that simple extraction tools fail more frequently in patients with high energy leads, necessitating powered mechanical sheaths and/or snares. Fortunately, procedural success and complication rates were still favourable in both groups, regardless of the technique used. The presence of a high energy lead as risk factor for a complex procedure has already been identified in the MB score [8]. Depending on inclusion criteria, the use of advanced extraction tools ranged from 52% to 100% in recent literature [16,17,20,22], similar to our cohort (59.2%). To the authors’ knowledge, no analysis of a high-volume centres evaluating patients with vs. without high energy leads has been published in the last five years, therefore incorporating today’s experience and tools. One of the latest high-number TLE analyses investigating major complications with vs. without high energy lead included data of 3258 patients until 2012 [24], showing no differences in the complication rate between both groups. Gould et al., recently evaluated the outcome of patients with vs. without cardiac resynchronization therapy (CRT) undergoing TLE, with similar outcomes in both groups [16].

The overall rate of acute complications was acceptable (4.0%) and remained stable during all five-year intervals. While all procedures were performed with cardiothoracic surgeons available on-site and ready for intervention, there was only one case of bail-out surgery within 667 procedures. Procedural and acute post-procedural mortality were also favourable (0.1% each). These outcomes are in line with previous studies, which showed a complete removal rate of 94–96% with a procedural major complication rate of 1.0–7.8% [15,17,18,19,20,21,22]. The results of our analysis showed lower complication rates than the multicentric ELECTRa study (with a major complication rate of 1.7% and death rate of 0.5%) [9], which may be explained by the high experience and strict standard operating procedure of our high-volume centre.

The MB score adequately identified patients in need of advanced extraction devices. Furthermore, we found the MB score correlates very well with the occurrence of short-term complications, although it was not validated for this outcome [8]. When evaluating single parameters, lead dwell time and a high number of implanted leads were predictive for the occurrence of acute complications. The modified SAFeTY TLE score was also correlated with both a complex procedure and procedural complications, despite the exclusion of the parameter “anaemia” due to missing data. Both scores may therefore help in clinical decision-making when the indication for TLE in a CIED patient is questioned.

### Limitations

This analysis is subject to bias due to its retrospective nature. The validation of the MB score for prediction of a complex procedure has to be taken with care as a part of the patients of this cohort has been used for development of the MB score. Furthermore, the SAFeTY TLE score could not be completely calculated due to missing data on anaemia. While this analysis adds valuable information on the acute outcome of TLE, long-term outcome data was, unfortunately, not available in this cohort.

## 5. Conclusions

This twenty-year experience proves that patients with high energy lead undergoing TLE require advanced extraction tools more often, but with a favourable outcome similar to comparator patients.

## Figures and Tables

**Table 1 jcm-11-04884-t001:** Device details and indications for TLE, stratified by ICD and non-ICD groups.

Parameter	Total (*n* = 667)	ICD (*n* = 405)	non-ICD (*n* = 262)	*p* Value
**Device details**				
device type				<0.001
- ICD	34.8% (*n* = 232)	52.6% (*n* = 213)	7.3% (*n* = 19)	
- CRT-D	33.7% (*n* = 225)	47.4% (*n* = 192)	12.6% (*n* = 33)	
- PM	29.2% (*n* = 195)	0% (*n* = 0)	74.4% (*n* = 195)	
- CRT-P	2.2% (*n* = 15)	0% (*n* = 0)	5.7% (*n* = 15)	
presence of a HIS lead	0.3% (*n* = 2)	0% (*n* = 0)	0.8% (*n* = 2)	0.154
indication for implant				<0.001
- DCM	31.3% (*n* = 209)	40.0% (*n* = 162)	17.9% (*n* = 47)	
- ICM	27.4% (*n* = 183)	37% (*n* = 150)	12.6% (*n* = 33)	
- AV block	17.1% (*n* = 114)	3.5% (*n* = 14)	38.2% (*n* = 100)	
- SSS	12% (*n* = 80)	1.5% (*n* = 6)	28.2% (*n* = 74)	
- inherited	8.4% (*n* = 56)	13.1% (*n* = 53)	1.1% (*n* = 3)	
- VT/VF	2.4% (*n* = 16)	3.0% (*n* = 12)	1.5% (*n* = 4)	
- not documented	1.3% (*n* = 9)	2.0% (*n* = 8)	0.4% (*n* = 1)	
implanted leads	2.4 ± 1.0	2.6 ± 0.9	2.1 ± 0.6	<0.001
**indication for TLE**				
infection	55.0% (*n* = 367)	61.7% (*n* = 250)	44.7% (*n* = 117)	<0.001
- pocket infection	35.1% (*n* = 234)	37.3% (*n* = 151)	31.7% (*n* = 83)	
- endocarditis	19.8% (*n* = 132)	24.4% (*n* = 99)	12.6% (*n* = 33)	
- not documented	0.7% (*n* = 5)	1.0% (*n* = 4)	0.4% (*n* = 1)	
lead dysfunction	39.4% (*n* = 263)	35.6% (*n* = 144)	45.4% (*n* = 119)	0.012
- lead failure	32.5% (*n* = 217)	28.9% (*n* = 117)	38.2% (*n* = 100)	
- lead interference	3.1% (*n* = 21)	0.7% (*n* = 3)	6.9% (*n* = 18)	
- recall (sprint fidelis)	3.4% (*n* = 23)	5.7% (*n* = 23)	0% (*n* = 0)	
- recall (riata)	0.1% (*n* = 1)	0.2% (*n* = 1)	0% (*n* = 0)	
- dislocation	0.1% (*n* = 1)	0% (*n* = 0)	0.4% (*n* = 1)	
other	6.4% (*n* = 43)	3.1% (*n* = 12)	11.3% (*n* = 31)	<0.001
- upgrade	3.4% (*n* = 23)	0.5% (*n* = 2)	7.7% (*n* = 21)	
- haematoma	0.1% (*n* = 1)	0.1% (*n* = 1)	0% (*n* = 0)	
- removal of lead stub	0.1% (*n* = 1)	0% (*n* = 0)	0.4% (*n* = 1)	
- not documented	2.7% (*n* = 18)	2.3% (*n* = 9)	3.3% (*n* = 9)	

Multiple indications for TLE were possible. TLE: transvenous lead extraction. ICD: implantable cardioverter defibrillator; CRT: cardiac resynchronization therapy; PM: pacemaker; DCM: dilated cardiomyopathy; ICM: ischemic cardiomyopathy; SSS: sick sinus syndrome; VT/VF: ventricular tachycardia/fibrillation.

**Table 2 jcm-11-04884-t002:** Lead details. RA: right atrium; RV: right ventricle; ICD: implantable cardioverter-defibrillator; LV: left ventricle.

	Total	ICD	non-ICD	*p* Value
**RA**, *n*	484	310	174	0.002
lead dwell time (months)	73.3 (38.9–111.4)	73.8 (40.9–109.0)	70.3 (32.0–119.4)	0.942
active fixation	28.5% (*n* = 138)	31% (*n* = 96)	24.1% (*n* = 42)	0.090
**RV (ICD)**, *n*	443	443	0	<0.001
lead dwell time (months)	59.1 (31.6–92.4)	59.1 (31.6–92.4)	N/A	N/A
dual coils	45.1% (*n* = 200)	45.1% (*n* = 200)	N/A	N/A
active fixation	49.4% (*n* = 219)	49.4% (*n* = 219)	N/A	N/A
**RV (PM)**, *n*	279	56	223	<0.001
lead dwell time (months)	77.0 (34.3–153.8)	120.4 (86.1–164.9)	62.45 (28.3–134.1)	<0.001
active fixation	31.5% (*n* = 88)	30.4% (*n* = 17)	31.8% (*n* = 71)	0.872
**LV**, *n*	223	181	42	<0.001
lead dwell time (months)	52.2 (28.9–82.5)	54.6 (28.9–84.4)	43.5 (28.4–69.5)	0.179
active fixation	2.7% (*n* = 6)	2.2% (*n* = 4)	4.8% (*n* = 2)	0.323
**Total**, *n*	1430	991	439	N/A
lead dwell time (months)	64.6 (33.9–104.1)	66.9 (35.6–100.1)	61.5 (30.1–109.4)	0.681
active fixation	31.5% (*n* = 451)	33.9% (*n* = 336)	26.2% (*n* = 115)	<0.001

RA: right atrium; RV: right ventricle; ICD: implantable cardioverter-defibrillator; LV: left ventricle.

**Table 3 jcm-11-04884-t003:** Extraction techniques used for successful lead explant.

Step	Total	ICD	non-ICD	*p* Value
**Step 1: Manual traction**	100% (*n* = 667)	100% (*n* = 405)	100% (*n* = 262)	N/A
Use of regular stylet	28.2% (*n* = 188)	27.9% (*n* = 113)	28.6% (*n* = 75)	0.860
Use of locking stylet	26.1% (*n* = 174)	26.9% (*n* = 109)	24.8% (*n* = 65)	0.588
Complete retrieval (step 1)	35.7% (*n* = 238)	22.5% (*n* = 91)	56.1% (*n* = 147)	<0.001
**Step 2: Non-powered dilator sheath**	2.4% (*n* = 16)	3.0% (*n* = 12)	1.5% (*n* = 4)	0.305
Complete retrieval (steps 1–2)	37.3% (*n* = 249)	24.7% (*n* = 100)	56.9% (*n* = 149)	<0.001
**Step 3: Powered sheaths**	59.2% (*n* = 395)	73.3% (*n* = 297)	37.4% (*n* = 98)	<0.001
Use of Mechanical sheaths	58.9% (*n* = 393)	73.3% (*n* = 297)	36.6% (*n* = 96)	<0.001
TightRail, Spectanetrics	6.1% (*n* = 41)	6.2% (*n* = 25)	6.1% (*n* = 16)	1.000
Evolution RL, Cook Medical	58.3% (*n* = 389)	72.6% (*n* = 294)	36.3% (*n* = 95)	<0.001
Use of Laser sheaths—SLS II, Spectanetrics	27.0% (*n* = 180)	33.3% (*n* = 135)	17.2% (*n* = 45)	<0.001
Complete retrieval (steps 1–3)	84.6% (*n* = 564)	84.7% (*n* = 343)	84.4% (*n* = 221)	1.000
**Step 4: Snare**	11.8% (*n* = 79)	12.6% (*n* = 51)	10.7% (*n* = 28)	0.540
Complete retrieval (steps 1–4)	94.8% (*n* = 632)	95.8% (*n* = 388)	93.1% (*n* = 244)	0.286
**Other approaches**	0.3% (*n* = 2)	0.2% (*n* = 1)	0.4% (*n* = 1)	1.000
Surgical removal	0.1% (*n* = 1)	0.0% (*n* = 0)	0.4% (*n* = 1)	0.391
Primary abandoning	0.1% (*n* = 1)	0.2% (*n* = 1)	0.0% (*n* = 0)	1.000
Complete retrieval	50.0% (*n* = 1)	0.0% (*n* = 0)	100.0% (*n* = 1)	1.000
**Overall success**				
Complete removal	94.9% (*n* = 633)	95.8% (*n* = 388)	93.5% (*n* = 245)	0.369
Clinical success	99.0% (*n* = 660)	98.8% (*n* = 400)	99.2% (*n* = 260)	0.256
Lead abandoned/failed TLE	1.0% (*n* = 7)	1.2% (*n* = 5)	0.8% (*n* = 2)	0.256
**Complications**	4.0% (*n* = 27)	4.7% (*n* = 19)	3.1% (*n* = 8)	0.421
Major intraprocedural complication	0.7% (*n* = 5)	0.7% (*n* = 3)	0.8% (*n* = 2)	1.000
Any intraprocedural complication	2.1% (*n* = 14)	2.2% (*n* = 9)	1.9% (*n* = 5)	1.000
Intraprocedural death	0.1% (*n* = 1)	0.2% (*n* = 1)	0% (*n* = 0)	1.000
Postprocedural complication	1.9% (*n* = 13)	2.5% (*n* = 10)	1.1% (*n* = 3)	0.267
Postprocedural death	0.1% (*n* = 1)	0.2% (*n* = 1)	0% (*n* = 0)	1.000

TLE: transvenous lead extraction.

**Table 4 jcm-11-04884-t004:** Predictors for short-term complications.

Parameter	Procedures with Complications	Procedures without Complications	*p* Value
**Single parameters**			
number of implanted leads	2.8 ± 1.2	2.4 ± 0.8	0.015
duration from lead implantation (years)	9.5 ± 6.3	6.7 ± 5.2	0.014
use of complex extraction tools (steps 2–3)	85.2%	60.9%	0.019
**Risk scores**			
MB score	4.5 ± 1.3	3.7 ± 1.4	0.005
SAFeTY TLE score	0.75 ± 0.64%	0.47 ± 0.46%	0.039

TLE: transvenous lead extraction.

## Data Availability

All data supporting reported results are available from the corresponding author upon reasonable request.

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
