# Peer review of "Step by Step through the Years—High vs. Low Energy Lead Extraction Using Advanced Extraction Techniques"

_jcm, 2022, doi:10.3390/jcm11164884_

Round 1
Reviewer 1 Report
I would like to thank the editors of JCM for the opportunity to review the manuscript by Zweiker et al.
In this retrospective study the authors compare the outcome after TLE of ICD vs. standard PM leads. The aim of the study is clear, the introduction is concise but would benefit at least of some more details on the most common risks during TLE. Although the used language is appropriate some sections lead to misunderstanding, therefore, the Reviewer recommends the involvement of a native speaker.
The authors should be congratulated on the impressive results of their daily work. In general, the manuscript is of high clinical relevance but could use some more structure and focus on the main outcome. I have the following comments
Why did the authors hypothesize that coils could be more dangerous for extraction?
Materials & Methods
The first paragraph materials and methods section sounds more like a PR text of San Raffaele, but should focus primarily on the methods of the study. How many extractions have been performed? The ethics approval ID should be added as well as the name of the review board, if available.
Was this a retrospective analysis? Please describe precisely.
In the Reviewer’s opinion: Study groups, outcomes, data collection and statistics should be mentioned prior the procedure.
Was the study registered at clinicaltrials.gov?
Stepwise TLE approach
Did all patients receive a temp pacing lead? What about patients who were not PM dependent?
Which cases were performed in the hybrid OR and which in the EP lab?
Who performed the cases? Were cardiac surgeons involved?
Data collection: which software was used?
Outcomes: What were the primary and the secondary outcomes for this study?
Results
It is not fully clear when the authors describe the results of all patients and when they compare the two groups. This section needs more structure.
Table 2: The authors show that in the RV (PM) subgroup 56 patients had an ICD lead. How is that possible?
Outcome: Again, the authors should start with the primary outcome and then report the other outcomes.
Is it true that 54.3% were extracted only by traction? Where is this represented in Table 3? The table shows 35.7% complete retrieval, which does not sum up with regular+locking stylet. Please clarify! How was the dwelling time of the manually extracted leads? How many leads were damaged with this technique?
Did the authors differentiate between single and dual coil ICD leads?
The authors should differ between major and minor complications (see Electra registry studies). 1 sternotomy and 1 periprocedural death in over 600 extractions are impressive results.
Discussion/Conclusion
The first paragraph in the discussion should summarize the main results of the study.
As far as the Reviewer understood the manuscript, the main aim is the comparison of outcome of ICD vs. non-ICD lead extaction. The ROC curves are an interesting information, but how do these provide relevant information for the main question? Therefore, these should probably be shown in a supplement.
As mentioned before the results are very impressive and significantly better (e.g. Death 0.1 vs. 1.2%, success rate 99.0% vs. 97%) than in the Electra Study. How do the authors explain this? This information would be extremely relevant for clinicians. And again, the incidence of major complications should be shown (Electra: 2.4%). What makes San Raffaele so different?
Minor details:
- High energy leads are defined for the first time in line 47, however, this should be done at first mentioning.
- Define all abbreviations when mentioned for the first time (e.g. Line 49)
- The tables do not have any legends
- Line 49: Reference for MB and SAFeTY TLE score are missing
- Line 60: referencs for guidelines are missing.
- Line 64: Did the authors mean lead rupture or “vein rupture” or “vessel tear”…?
- Table 1: the variables should be sorted: e.g. it is not clear from the beginning that infection = pocket infection + endocarditis
- Table 2: title: non-ICD instead of no-ICD.
Author Response
I would like to thank the editors of JCM for the opportunity to review the manuscript by Zweiker et al.
In this retrospective study the authors compare the outcome after TLE of ICD vs. standard PM leads. The aim of the study is clear, the introduction is concise but would benefit at least of some more details on the most common risks during TLE. Although the used language is appropriate some sections lead to misunderstanding, therefore, the Reviewer recommends the involvement of a native speaker.
The authors should be congratulated on the impressive results of their daily work. In general, the manuscript is of high clinical relevance but could use some more structure and focus on the main outcome. I have the following comments
- We thank the reviewer for his/her comments. We went through the whole manuscript with a native speaker and made several corrections. We adapted the introduction and included details on the most common risks during TLE.
Why did the authors hypothesize that coils could be more dangerous for extraction?
- Segreti et al (Heart Rhythm 2014, DOI 10.1016/j.hrthm.2014.08.011) found an association between coils and lead adhesions in TLE patients. We added this reference to the introduction (Reference [5]).
Materials & Methods
The first paragraph materials and methods section sounds more like a PR text of San Raffaele, but should focus primarily on the methods of the study. How many extractions have been performed?
- We agree with the reviewer and removed the first sentence of the Methods. Furthermore, we added the mean number of TLE procedures per year in the second sentence.
The ethics approval ID should be added as well as the name of the review board, if available.
- The institutional review board (Vita-Salute University San Raffaele) approved the study without formal application because it represents a strictly retrospective analysis of anonymized data. We added the name of the institutional review board.
Was this a retrospective analysis? Please describe precisely.
- The data was collected prospectively, but the analysis of consecutive patients was retrospective. We added “This is a retrospective analysis […]” into the Methods section.
In the Reviewer’s opinion: Study groups, outcomes, data collection and statistics should be mentioned prior the procedure.
- As this study is a retrospective analysis, we went for a chronological description of the analysis and therefore put the procedural details first.
Was the study registered at clinicaltrials.gov?
Stepwise TLE approach
Did all patients receive a temp pacing lead? What about patients who were not PM dependent?
- As per our standard operating procedure, all patients received a temporary pacing lead (a standard quadripolar EP catheter) for the procedure. We have seen complete AV block as acute complication in TLE. We therefore introduced this step even in patients without PM dependence.
Which cases were performed in the hybrid OR and which in the EP lab?
- All TLE procedures were performed in the electrophysiology laboratory, as written in the first sentence of the section “Stepwise approach for TLE”. However, the extraction itself was only performed with a cardiac surgery team available for immediate cardiac surgery. In case of a complication, the patient was either immediately transferred into a cardiac surgery theatre. We added “[…] with a cardiac surgery team ready” to the first sentence.
Who performed the cases? Were cardiac surgeons involved?
- Experienced electrophysiologists performed the procedures, but cardiac surgeons were available in case of any severe complication. We added “[…] with a cardiac surgery team ready” to the first sentence of the section “Stepwise approach for TLE”.
Data collection: which software was used?
- We used Microsoft Excel for data collection. We added this information to the section “Data collection”.
Outcomes: What were the primary and the secondary outcomes for this study?
- The primary outcome was “clinical success” and the secondary outcomes were “use of advanced extraction tools” and “any complications”. We added this information to the section “Outcome”.
Results
It is not fully clear when the authors describe the results of all patients and when they compare the two groups. This section needs more structure.
- We tried to describe the results chronologically from indication to the procedure and outcome. However, we adapted this section and now concentrate on the difference between both groups.
- During the rewriting of this section, we found an error of the total number of patients in ICD vs. no-ICD groups. We excluded further errors (except for minor rounding errors) by checking all parameters in all tables. We therefore had to make several changes in the abstract, the graphical abstract, the results section, Table 1 and Table 3.
Table 2: The authors show that in the RV (PM) subgroup 56 patients had an ICD lead. How is that possible?
- The situation is vice-versa: Those 56 patients in the ICD group had an additional RV-PM lead that was extracted. The majority of those patients previously had a PM and then got an upgrade to ICD without removal of the PM lead.
- However, there actually were patients with ICD in the ICD group. Those patients had an ICD in place, but the ICD lead was not extracted. Most of these patients either had a dysfunctional PM lead or received TLE of an abandoned RV pacemaker lead before upgrade to CRT.
Outcome: Again, the authors should start with the primary outcome and then report the other outcomes.
- We adapted the “Outcome” section, first presenting the main outcomes (clinical success, use of advanced extraction tools and any complications).
Is it true that 54.3% were extracted only by traction? Where is this represented in Table 3? The table shows 35.7% complete retrieval, which does not sum up with regular+locking stylet. Please clarify! How was the dwelling time of the manually extracted leads? How many leads were damaged with this technique?
- We acknowledge that Table 3 is tricky to read but we did not come up with a better solution to show the distinct steps of our stepwise approach. Manual traction was tried in all patients. Regular and locking stylets were only used in a fraction of patients. The results of each step can be read in the “complete retrieval” lines: Manual traction was successful in only 35.7% of procedures. We tried to clarify the table to add “use of […]” at the beginning of many lines.
- We did not look into dwelling time of manually extracted leads, because we already published a detailed analysis on the predictors of the use of advanced extraction methods: Bontempi, L., et al. (2020). "The MB score: a new risk stratification index to predict the need for advanced tools in lead extraction procedures." Europace 22(4): 613-621.
Unfortunately, we do not have the exact information how the leads were damaged during manual traction. We only have information, in which procedures the operators proceeded with advanced extraction techniques.Did the authors differentiate between single and dual coil ICD leads?
- To avoid overcomplication of this analysis, we decided not to differentiate between singly and dual coil ICD leads.
The authors should differ between major and minor complications (see Electra registry studies). 1 sternotomy and 1 periprocedural death in over 600 extractions are impressive results.
- We added the term “major complication”, defined it in the Methods section and displayed the results in Table 3. There were no significant differences between both groups in terms of major complications (ICD-group: 0.7% vs. no-ICD-group 0.8%)
Discussion/Conclusion
The first paragraph in the discussion should summarize the main results of the study.
- We hope that the first paragraph of our manuscript does summarize the main results of the study. We changed the tense and added the total number of TLE procedures.
As far as the Reviewer understood the manuscript, the main aim is the comparison of outcome of ICD vs. non-ICD lead extaction. The ROC curves are an interesting information, but how do these provide relevant information for the main question? Therefore, these should probably be shown in a supplement.
- We agree and put the ROC curves into the supplement.
As mentioned before the results are very impressive and significantly better (e.g. Death 0.1 vs. 1.2%, success rate 99.0% vs. 97%) than in the Electra Study. How do the authors explain this? This information would be extremely relevant for clinicians. And again, the incidence of major complications should be shown (Electra: 2.4%). What makes San Raffaele so different?
- We found a major complication rate of 1.7% in the Electra Study (Eur Heart J 38(40): 2995-3005). The high experience of our centre, combined with a strict standard operating procedure incorporating the stepwise approach may play a role in this difference. We added a paragraph trying to explain those differences in the Discussion section.
Minor details:
High energy leads are defined for the first time in line 47, however, this should be done at first mentioning.
- We moved the definition up.
Define all abbreviations when mentioned for the first time (e.g. Line 49)
- We checked the manuscript for abbreviations and added definitions, where applicable.
The tables do not have any legends
- We added legends (footers) to the tables and moved abbreviation definitions there, if applicable.
Line 49: Reference for MB and SAFeTY TLE score are missing
- To put those risk scores more into background, we chose to remove the names from the Introduction and referenced them accordingly in the Methods section.
Line 60: referencs for guidelines are missing.
- We added a reference to the sentence.
Line 64: Did the authors mean lead rupture or “vein rupture” or “vessel tear”…?
- We meant lead rupture. In our centre, the snare is mostly used as “bailout” in case of a ruptured lead.
Table 1: the variables should be sorted: e.g. it is not clear from the beginning that infection = pocket infection + endocarditis
- We added hyphens to sub-parameters for clarification.
Table 2: title: non-ICD instead of no-ICD.
- We thank the reviewer for this comment and now use “non-ICD” throughout the manuscript.
Reviewer 2 Report
This manuscript summarized transvenous lead extraction over 20 years with a clear step-wise approach of explantation techniques. As limited data is available according to this topic, there is a need of publication of these data. The manuscript is innovative, interesting, and well-written.
Line 141: Out of 1430 EXPLANTED leads
Line 143: Indicated IQR for median lead dwell times
Line 158/159: Indicated absolute numbers for 58.7% and 10.5% and 57.2%.
Line 189-191: Indicated whether these complications occurred in the ICD or non-ICD group
Line 206: If possible, a subgroup analysis of the modified SAFeTY TLE score should be made in patients with available anaemia-data in the data supplement.
Line 244: Remove one “had”
Line 287: Add the limitations of a single-center retrospective study and (if still indicated) the limitation of the modified SAFeTY TLE score without the data of “anaemia”.
Congratulations!
Author Response
This manuscript summarized transvenous lead extraction over 20 years with a clear step-wise approach of explantation techniques. As limited data is available according to this topic, there is a need of publication of these data. The manuscript is innovative, interesting, and well-written.
- We thank the reviewer for this comment.
Line 141: Out of 1430 EXPLANTED leads
- These details refer to the implanted leads in treated patients.
Line 143: Indicated IQR for median lead dwell times
- We thank the reviewer for this comment and we added IQR.
Line 158/159: Indicated absolute numbers for 58.7% and 10.5% and 57.2%.
- We added absolute numbers.
Line 189-191: Indicated whether these complications occurred in the ICD or non-ICD group
- We added this information. Furthermore, we added another parameter “major complications” similar to the ELECTRa registry manuscript.
Line 206: If possible, a subgroup analysis of the modified SAFeTY TLE score should be made in patients with available anaemia-data in the data supplement.
- Unfortunately, we have anaemia-data only in <5% of cases. Therefore, we think that it is not reasonable.
Line 244: Remove one “had”
- We removed one “had”.
Line 287: Add the limitations of a single-center retrospective study and (if still indicated) the limitation of the modified SAFeTY TLE score without the data of “anaemia”.
- We added those limitations.
Congratulations!
- Thank you.
- We have to acknowledge that during rewriting of the results section we found an error of the total number of patients in ICD vs. no-ICD groups. We excluded further errors (except for minor rounding errors) by checking all parameters in all tables. We therefore had to make several changes in the abstract, the graphical abstract, the results section, Table 1 and Table 3.
Round 2
Reviewer 1 Report
The authors answered all my questions, I have no further remarks.